# Homing and Engraftment of Intravenously Administered Equine Cord Blood-Derived Multipotent Mesenchymal Stromal Cells to Surgically Created Cutaneous Wound in Horses: A Pilot Project

**DOI:** 10.3390/cells9051162

**Published:** 2020-05-08

**Authors:** Suzanne J. K. Mund, Eiko Kawamura, Awang Hazmi Awang-Junaidi, John Campbell, Bruce Wobeser, Daniel J. MacPhee, Ali Honaramooz, Spencer Barber

**Affiliations:** 1Department of Large Animal Clinical Sciences, Western College of Veterinary Medicine, University of Saskatchewan, Saskatoon, SK S7N 5B4, Canada; john.campbell@usask.ca (J.C.); spence.barber@usask.ca (S.B.); 2WCVM Imaging Centre, Western College of Veterinary Medicine, University of Saskatchewan, Saskatoon, SK S7N 5B4, Canada; eiko.kawamura@usask.ca; 3Department of Veterinary Biomedical Sciences, Western College of Veterinary Medicine, University of Saskatchewan, Saskatoon, SK S7N 5B4, Canada; awanghazmi@upm.edu.my (A.H.A.-J.); d.macphee@usask.ca (D.J.M.); ali.honaramooz@usask.ca (A.H.); 4Department of Veterinary Pathology, Western College of Veterinary Medicine, University of Saskatchewan, Saskatoon, SK S7N 5B4, Canada; bruce.wobeser@usask.ca

**Keywords:** stem cells, equine limb wound, exuberant granulation tissue, intravenous, horse

## Abstract

Limb wounds on horses are often slow to heal and are prone to developing exuberant granulation tissue (EGT) and close primarily through epithelialization, which results in a cosmetically inferior and non-durable repair. In contrast, wounds on the body heal rapidly and primarily through contraction and rarely develop EGT. Intravenous (IV) multipotent mesenchymal stromal cells (MSCs) are promising. They home and engraft to cutaneous wounds and promote healing in laboratory animals, but this has not been demonstrated in horses. Furthermore, the clinical safety of administering >1.00 × 10^8^ allogeneic MSCs IV to a horse has not been determined. A proof-of-principle pilot project was performed with two horses that were administered 1.02 × 10^8^ fluorescently labeled allogeneic cord blood-derived MSCs (CB-MSCs) following wound creation on the forelimb and thorax. Wounds and contralateral non-wounded skin were sequentially biopsied on days 0, 1, 2, 7, 14, and 33 and evaluated with confocal microscopy to determine presence of homing and engraftment. Results confirmed preferential homing and engraftment to wounds with persistence of CB-MSCs at 33 days following wound creation, without clinically adverse reactions to the infusion. The absence of overt adverse reactions allows further studies to determine effects of IV CB-MSCs on equine wound healing.

## 1. Introduction

Cutaneous wounds are common in horses and often must heal by second intention. Compared to body wounds, limb wounds have prolonged low-grade inflammation which causes slower healing and less contraction, and this in turn promotes development of exuberant granulation tissue (EGT) and a less durable and less cosmetic repair [1,2]. Keloid formation in humans is similar to EGT formation on the limbs of horses and, interestingly, humans and horses are the only two species that spontaneously develop these fibroproliferative disorders, making the horse a good animal model for studying keloids and hypertrophic scars in humans [3].

Multipotent mesenchymal stromal cells (MSCs) have been investigated as ancillary therapy for complicated cutaneous healing and keloid treatment in humans [4,5,6,7]. MSCs improve wound healing by modulating the immune response both systemically and locally [8,9,10] and these positive effects are further enhanced when MSCs successfully home to and engraft within the wound [11]. However, experimental studies examining the effects of multipotent cells on equine cutaneous wounds are limited [12,13,14].

In horses, autologous MSCs are most commonly used for tendon healing and a lag time is required for isolation and expansion of MSCs before administration [15,16]. While the most appropriate time for administration of MSCs to enhance healing of cutaneous wounds is currently unknown, in laboratory animal models of spinal cord contusion [17] and colitis [18], MSC therapy during the acute inflammatory phase is crucial to best ameliorate the inflammatory response. Thus, delaying therapy to culture autologous MSCs may miss the “window of opportunity” for a good outcome. Allogeneic MSCs (allo-MSCs) can be isolated and expanded prior to injury for timely administration [16], and intravenous (IV) allo-MSCs have been shown to positively influence cutaneous wound healing in several species [7,9] but this effect has not yet been demonstrated in horses.

In human regenerative medicine, there is great interest in umbilical cord blood-derived MSCs (CB-MSCs) as collection is non-invasive, they can be stored long-term for future autologous or allogeneic use, are immunomodulatory and have higher expression of pluripotency markers as well as higher isolation and proliferation rates compared to bone marrow-derived MSCs [19]. Equine CB-MSCs are a very promising source of allo-MSCs because they can be similarly collected, stored, and expanded [20] and have been shown to have immunomodulatory properties in vitro [21,22].

MSCs may be administered to cutaneous wounds locally (i.e., topically [12,23] or intradermally [12,24,25]), systemically [10,26,27] or via regional limb perfusion [28,29]. Although local administration is intuitive, complications such as a tissue damage from the needle tract and local compartmentalization compromising blood supply can occur [10,28,30]. Furthermore, local administration does not always improve engraftment compared to systemic methods and its beneficial effects are appreciated only locally rather than systemically [10]. While systemic administration of MSCs is minimally invasive and convenient, complications such as pulmonary injury [31,32], cerebral embolism [33] and hypersensitivity responses [34] are possible. Safety studies of relatively small amounts of IV administered MSCs (0.2–50 × 10^6^) [35,36,37,38] have been performed in horses but to our knowledge no equine studies have investigated whether there are complications associated with IV administration of >1.0 × 10^8^ MSCs, or whether IV administered MSCs can home and engraft into equine cutaneous wounds.

The primary objective of this proof-of-principle pilot study was to determine if subject horses develop adverse reactions during or for 6 weeks post-IV administration of >1.0 × 10^8^ fluorescently prepared allo-CB-MSCs. The secondary objectives were to describe the presence and patterns of homing and engraftment of fluorescently prepared allo-CB-MSCs in biopsies from wounded and non-wounded cutaneous tissue of the limb and thorax during healing, and to determine whether preliminary results suggest limb wounds have different patterns of CB-MSC homing and engraftment than thoracic wounds over time.

## 2. Materials and Methods

The experimental protocols (AUP #20140096) involving animals were approved by the University Animal Care Committee and Animal Research Ethics Board of the University of Saskatchewan. 

### 2.1. Overview of the Study Design

Standardized cutaneous wounds were created on the left lateral third metacarpus (MCIII) and left hemi-thorax. Twelve hours after wound creation, 1.02 × 10^8^ live allogeneic CB-MSCs labeled with a fluorescent dye (PKH26) and transduced with an enhanced green fluorescent protein (eGFP) transgene were administered intravenously via the jugular vein. Red fluorescence of PKH26 would allow immediate identification of CB-MSCs in biopsy samples before green fluorescence of translated eGFP was consistently produced by viable CB-MSCs. Detection of red fluorescent signal of CB-MSCs in the early stages of wound healing was considered indicative of homing but not necessarily viability and engraftment, whereas detection of green and red fluorescent signal of CB-MSCs outside of the vasculature was considered indicative of homing, transendothelial migration, and engraftment of viable CB-MSCs. Biopsies were taken of wounds on the limbs and thoraces and of the corresponding contralateral non-wounded control sides on days 0, 1, 2, 7, 14, and 33 and evaluated by confocal microscopy for presence of administered prepared CB-MSCs. Horses were monitored for adverse clinical reactions during injection and for the duration of the study period of 6 weeks.

### 2.2. Recipient Animals

Two sound 7-year-old Thoroughbred mares (average weight 460 kg; 485 kg and 435 kg each) with no history of injury to the forelimbs or thoracic wall that were healthy on physical exam with normal complete blood counts and serum chemistry were used. Horses were vaccinated (West Nile-Innovator+EWT, Zoetis Canada, Inc., Kirkland, QC, Canada) and dewormed (Equimax, Bimeda Canada, Cambridge, ON, Canada). 

### 2.3. Transducing and Labeling Trial

Prior to commencement of the project, a trial sample of 1.0 × 10^6^ CB-MSCs (eQcell Therapies Inc.; King City, ON, Canada) was acquired in order to determine logistics and timing of shipping and receiving, and to establish ideal transducing and labeling conditions. CB-MSCs were of the same source as the CB-MSCs used in the in vivo project. Optimal multiplicity of infection (MOI) and concentration of PKH26 were determined using the same AAV2-CMV-eGFP vector (AAV2.CMV.PI.eGFP.WPRE.bGH; Penn Vector Core, Gene Therapy Program, University of Pennsylvania) and PKH26 labeling kit (PKH26 Red Fluorescent Cell Linker; Sigma-Aldrich, Oakville, ON, Canada) as in the in vivo project. After transducing and labeling, CB-MSCs were cultured long term to determine fluorescent expression patterns of prepared CB-MSCs, establish label longevity, verify passage of fluorescence to daughter cells, and to confirm absence of cell toxicity. Using a similar protocol, we have previously shown that almost all testis stem cells exposed to AAV2 were successfully transduced with eGFP [39] and stably expressed the eGFP transgene for several years in vivo [40]. Additionally, by optimizing the PKH26 labeling protocol we were able to maximize detection of red fluorescence in live transplanted germ cells for up to 12 weeks [41,42].

### 2.4. Source and Transportation of CB-MSCs

CB-MSCs originating from cord blood of five unrelated male donor foals were used (eQcell Therapies Inc.). This source of CB-MSCS has been previously characterized and demonstrated to be capable of trilineage differentiation [20] and expression of a surface marker phenotype consistent with MSCs [22]. The donor cells were isolated, cultured, and processed using procedures that have been previously described [20]. At the time of shipment, the CB-MSCs had been cryopreserved once, passaged 4–5 times, and cultured for a total of 35–45 days. Twenty-four hours prior to injection, they were harvested using trypsin and disodium ethylenediaminetetraacetic acid (EDTA), washed in phosphate buffered saline (PBS), pooled to obtain a mixed population of donor cells, re-suspended in a serum-free commercial cell preservation medium (HypoThermosol FRS [HTS-FRS]; BioLife Solutions, Bothell, WA, USA) and cooled. The cooled CB-MSCs were shipped overnight to our laboratory in a reusable temperature-controlled shipping container (Greenbox 2–8 °C thermal management system, ThermoSafe, Arlington Heights, IL, USA). A total of 4.16 × 10^8^ cooled CB-MSCs were provided as it was the maximum number of CB-MSCs that the supplier could produce at one time with their facilities and resources. A representative sample was taken and cell count and viability determined in duplicate and averaged using a haemocytometer counting chamber and trypan blue exclusion assay. The cell viability was assessed to be 80%, resulting in 3.33 × 10^8^ total live cells received.

### 2.5. Preparation of CB-MSCs

Donor CB-MSCs were transduced in vitro with an eGFP transgene via an AAV2-CMV-eGFP vector as described in Section 2.3. CB-MSCs were exposed to the AAV2 at MOI of 1.0 × 10^4^ genome copies/cell. CB-MSCs were then incubated at 37 °C for 3 h in vitro, washed twice by centrifugation at 500× *g* for 5 min, and re-suspended in Dulbecco modified Eagle medium (DMEM; Mediatech, Manassas, VA, USA). 

After transduction, CB-MSCs were labeled by incubation with PKH26 (Sigma-Aldrich) as previously described [41,42]. Briefly, CB-MSCs were washed in DMEM, centrifuged at 400× *g* for 5 min, and re-suspended in Diluent C. Immediately before staining, 2.0 × 10^−5^ molar of PKH26 dye was prepared using Diluent C, gently mixed with the CB-MSCs, and incubated at 25 °C for 6 min. Staining was stopped by the addition of fetal bovine serum (PAA Laboratories, Etobicoke, ON, Canada), and CB-MSCs were subsequently washed three times in DMEM. After the last wash, CB-MSCs were re-suspended in HTS-FRS for a total volume of 60 mL in a sterile syringe and kept cool until injection one hour later. After preparation, a representative sample of CB-MSCs was retained and cell count and viability were repeated using a haemocytometer counting chamber and trypan blue exclusion assay. There was a total of 2.04 × 10^8^ live CB-MSCs available and hence 1.02 × 10^8^ live CB-MSCs were administered to each horse.

Additionally, a sample of prepared CB-MSCs was cultured into 6-well plates to serve as an in vitro reference for timing and pattern of fluorescence of prepared CB-MSCs. The supernatant from the last cell wash was also added to a cell culture of a sample of non-prepared CB-MSCs to confirm no contamination of the supernatant with free PKH26. These cultures were maintained and observed for the duration of the study.

### 2.6. Wound Creation

Twelve hours prior to surgery, an intravenous catheter was aseptically placed in the left jugular vein and feed was restricted eight hours pre-operatively. On day 0, horses were anaesthetized, maintained to effect on a guaifenesin, ketamine and xylazine intravenous drip (1 L 5% guaifenesin + 1000 mg ketamine + 500 mg xylazine) and placed in right lateral recumbency. After aseptic preparation, seven standardized full thickness excisional skin wounds were created using a scalpel on the left lateral MCIII and hemi-thorax at the region of the tenth costochondral junction of each horse (Figure 1). Wounds measured 0.5 cm × 2.0 cm in a horizontal orientation and orientated in a vertically stacked arrangement 2.0 cm apart. The wounds were covered during recovery from anesthesia and then were left unbandaged to heal by second intention. Excised skin was retained for evaluation of baseline background fluorescence. Anti-inflammatories and antimicrobials were not administered at any time to avoid modification of inflammation.

### 2.7. Prepared CB-MSC Administration and Monitoring

On day 1, twelve hours after wound creation, the prepared CB-MSCs were injected via the indwelling catheter (4 mL/min over 15 min). During the injection, vital parameters were monitored for adverse clinical reactions (i.e., tachycardia, tachypnea, pyrexia, respiratory distress, colic, urticaria) every minute for the first 5 min followed by every 5 min until the suspension was administered. A physical exam was performed every 12 h for the following 36 h then once daily for the following 7 days, followed by distance exam (i.e., demeanor, lameness at the walk, excessive swelling or discharge from wounds) until the conclusion of the study 6 weeks after injection.

### 2.8. Biopsy Collection

The two most proximal wounds on the limb and thorax were not biopsied and observed for healing characteristics in preparation for future studies. Biopsies were taken of the remaining surgically created wounds of the limbs and thoraces and of the corresponding contralateral control sides on days 1, 2, 7, 14, and 33 in a distal to proximal sequence (Figure 1). During biopsy collection, horses were sedated (0.3 mg/kg xylazine + 0.01 mg/kg butorphanol) and ~3 mL of 2% lidocaine was injected subcutaneously at the biopsy site of interest. On the wounded side, a full thickness skin biopsy consisting of ~3 mm of the granulating wound and 3 mm of adjacent skin the entire length of the original wound (~20 mm) was excised using a #11 scalpel. On the contralateral non-wounded side, a template was used to trace an outline of 0.5 × 2.0 cm using a sterile surgical marker and a full thickness biopsy excised using a #11 scalpel. Biopsied tissue was cooled and immediately processed and evaluated by confocal microscopy.

### 2.9. Fluorescence Evaluation of Biopsies

Confocal microscopy was used to determine normal background fluorescence of resected skin collected on day 0 and to detect fluorescent signal in biopsy tissue from administered prepared CB-MSCs. One mm slices of fresh cooled biopsies were prepared and examined under a Leica epifluorescence microscope for red and green fluorescent signal using filter sets for GFP (excitation 450–490 nm/emission LP515 nm) and rhodamine (excitation 515–560 nm/emission LP590 nm) with a 63× oil objective and imaged with a Leica SP5 confocal microscope (Leica Microsystems; Watzlar, Germany) using the 488 nm line of an argon laser (emission range of 500–535 nm for GFP) and the 543 nm line of a helium neon laser (emission range of 555–700 nm for PKH26). Biopsies from each collection period were evaluated by a blinded operator (E.K.). The operator subjectively evaluated the biopsies for presence of red and/or green signal and the intensity and distribution of the signal. Fluorescing cell-like structures were considered administered CB-MSCs if they had similar fluorescence patterns and were of similar size to cultured CB-MSCs in our in vitro study. Fluorescent signal patterns were thoroughly described for each horse and then summarized. No difference in homing between the wound biopsies and the contralateral non-wounded biopsies was recorded as =, slightly more homing to wounds was recorded as +1, moderately more homing was recorded as +2, and markedly more homing was recorded as +3. Presence of red fluorescence was recorded as *R* and presence of green fluorescence was recorded as *G*. Biopsies in which signal was absent or rarely detected was further recorded as *none* or *rare*.

## 3. Results

### 3.1. Adverse Reactions

Neither of the horses developed adverse clinical reactions during injection of the prepared CB-MSCs and all evaluated vital parameters remained within normal limits. One horse had symptoms of mild colic (abdominal pain) prior to injection of the CB-MSCs that was attributed to anesthesia and because her vital signs remained normal the CB-MSCs were administered. After administration of the CB-MSCs, she responded well to minimal intervention (oral fluids, oral mineral oil, sedation [150 mg xylazine + 5 mg butorphanol IV]) and did not experience any more colic symptoms throughout the duration of the study period. There was mild edema of the tissue adjacent to the wounds in both horses for a few days following wound creation and biopsy collection; however, lameness was not apparent at the walk and intervention was not needed. Both horses remained comfortable and did not develop any adverse reactions throughout the remainder of the study period. 

### 3.2. Confocal Microscopy Observations

#### 3.2.1. In Vitro Culture

On all days, CB-MSCs measured ~15–20 μm in length and 10–15 μm in width. On day 1, prepared CB-MSCs had dots of predominantly red fluorescence measuring on average 2 μm uniformly distributed throughout the cell on phospholipid membranes and very weak green signal was also detected as dots (Figure 2a). On day 7, dots of red fluorescence remained uniformly distributed and weak green signal was detected as small dots <1 μm in diameter and/or generalized green fluorescence of the cytoplasm with low amounts of colocalization of green and red signal (Figure 2b). On days 14 and 33, the distribution of green signal remained similar to day 7 but more vibrant and readily detectable (Figure 2c,d). Green signal was most intense on day 33 (Figure 2d). On day 14 and 33, the amount of colocalization of green and red signal varied between individual cells but was typically low to moderate.

#### 3.2.2. In Vivo Tissue Biopsies

##### Day 0 

Collagen fibers demonstrated normal green and red auto-fluorescence consistently in limb and thoracic biopsies of both horses. There were no cell-like structures that had similar fluorescent patterns to cultured prepared CB-MSCs (Figure 3). 

##### Day 1 

Red and green signals were typically colocalized and detected in all biopsies except in the non-wounded thoracic biopsy of horse 1. Signal was typically highly concentrated in the vasculature and both horses had slightly more signal in the wounded limb and thoracic biopsies (Table 1; Figure 4a,c). 

##### Day 2 

Colocalized red and green signal was detected in all biopsies. In biopsies of limb wounds of both horses, slightly more signal was detected compared to non-wounded limb biopsies. There was no difference in amount of signal or pattern of distribution in wounded and non-wounded thoracic biopsies of both horses. In all biopsies, signal was detected within the vasculature in clusters, but also was located along the endothelium of the vessels, possibly consistent with endothelial adhesion of CB-MSCs (Figure 5c,d). In wounded and non-wounded limb biopsies of horse 1, cell-like structures measuring ~8–10 μm were occasionally detected outside of the vasculature in the interstitium (Table 1; Figure 5b).

##### Day 7 

Red and green colocalized signal was detected in all biopsy sites of both horses. In horse 1, there was no difference in amount of detected signal from wounded or non-wounded biopsies of the limb or thorax. Furthermore, in both wounded and non-wounded limb and thoracic sites, signal was detected primarily within and along the endothelium of the vasculature (Figure 6c,d). In wounded and non-wounded limb biopsies, occasional cell-like structures were detected in the interstitium measuring 8–10 μm (Figure 6a) and larger cell-like structures measuring 15–20 μm were detected within the vasculature along the endothelium (Figure 6b). In horse 2, there was no difference in detected signal in wounded and non-wounded limb biopsies. Occasional cell-like structures measuring 12–20 μm were detected within the vasculature of both wounded and non-wounded limb sites (Figure 6a). In the thorax, there was one area in the wounded site that contained several cell-like structures measuring 8–11 μm in the interstitium, but otherwise there was no difference in detected signal between wounded and non-wounded biopsies. Cell-like structures were occasionally detected within the vasculature of both wounded and non-wounded thoracic sites (Table 1). 

##### Day 14 

In both horses, there was no difference in pattern of distribution of red or green signal, or intensity of fluorescence in limb or thoracic wounded and non-wounded biopsies. When present, red and green signal was typically colocalized and located both within and out of the vasculature. In addition, signal that formed clumps of ~5 μm was often found along the endothelial surface of vessels, just adjacent to the vasculature or in the interstitium (Figure 7a). Cell-like structures were rarely identified; a single cell-like structure measuring 15 μm was detected in the interstitium of the non-wounded limb of horse 2 and wounded thorax of horse 1 (Table 1; Figure 7b,c).

##### Day 33 

On day 33: red and green signal was more readily apparent in limb and thoracic wounds of both horses (Figure 8a,c). More specifically, signal was colocalized and predominately detected as cell-like structures between 10–20 μm (Table 1). 

These cell-like structures were found in clusters and larger cell-like structures measuring 15–20 μm were often elongated. Both larger and smaller cell-like structures appeared to be integrated into the interstitium. In the non-wounded limb and thoracic biopsies, normal background fluorescence was typical and cell-like structures that were rarely detected were rounded, located often within the vasculature, and measured between 5–12 μm, smaller than those typically detected in the wounded biopsies (Figure 8b).

## 4. Discussion

Limb wounds in horses produce a chronic low-grade inflammatory stimulus that causes dysregulation of cytokines which delays contraction, epithelialization, and angiogenesis, and stimulates formation of exuberant granulation tissue [43,44,45,46,47,48,49,50]. Similarly, dysregulated cytokine release can also cause delayed healing and hypertrophic scar and keloid formation in chronic wounds in people [7,51,52,53,54]. Compared to body wounds, acute equine limb wounds have a less robust cellular inflammatory response which is inadequate to trigger the local wound environment to transition from proinflammatory to one that promotes resolution of inflammation and remodeling [1,2]. Systemically administered MSCs promote healing of a chronic wound by homing to the wound and downregulating release of proinflammatory mediators while upregulating release of anti-inflammatory mediators, and promote differentiation of fibroblasts to myofibroblasts to assist in contraction [54]. Clinical trials of MSC therapy in humans with chronic wounds have had favorable outcomes [7]. In the equine literature, only a single experimental study [12] and few case reports [55,56,57] using topical MSCs to promote healing of limb wounds have been performed, with promising results [12]. Intravenous MSC therapy has great promise to prevent compromised healing in horse limb wounds and resolve chronic low-grade inflammation of chronic wounds.

To the authors’ knowledge, 1.02 × 10^8^ is the largest amount of allo-MSCs that have been administered intravenously to horses, and we did not detect any clinical adverse reactions directly attributable to allo-MSC therapy. This is encouraging because despite evidence that equine MSCs have immunomodulatory properties [22,58,59,60,61,62,63,64], and intravenously administered MSCs are well tolerated [35,36,37,38], complications and reactions secondary to allo-MSC therapy and IV MSC administration can occur. Previously reported immune responses in horses to allo-MSCs include allo-antibody production [34,37,65], increased circulating lymphocyte population [36,38], wheal formation and local lymphocyte response after intradermal administration [13,34,66], and increased lameness and nucleated cell counts after intrasynovial injection [67]. Regardless, in these studies all local reactions were self-limiting and no horses developed systemic adverse reactions. Reported complications related to intravascular MSC administration in other species include fatal pulmonary embolism in mice [31], cerebral embolism in rats [33], pulmonary parenchymal edema and hemorrhage in dogs [32], and portal vein thrombosis in humans [68]. Despite these reports, complications are typically not encountered and IV administration of MSCs appears to be well-tolerated in most species [69,70]. However, laboratory animals used in IV MSC studies are often administered upwards of 1.6 × 10^7^ MSCs per kg [10,71] and the minimal effective IV dose in humans for a variety of conditions, including neurologic conditions and graft versus host disease, is 7.00 × 10^7^ to 1.90 × 10^8^ MSCs per patient, or 1.0 × 10^6^ to 2.7 × 10^6^ MSCs per kg for the average 70 kg human patient [72]. Our horses received a comparatively modest dose of 2.22 × 10^5^ MSCs per kg, the largest dose that our supplier could provide. This is 1/5th to 1/12th of the dose commonly administered to humans, and 1/72nd of the dose typically administered to laboratory animals. Clearly, administering similar amounts of MSCs per kg to a horse would be cost prohibitive and very technically challenging to produce such a high dose. However, while the ideal dose of MSCs for a beneficial effect is unknown, neither positive clinical effects nor adverse reactions may be seen unless higher doses than previously administered amounts are used; therefore, we felt it important to evaluate the effects of a larger dose of IV MSCs than had been previously administered. Although we did not evaluate peripheral blood or tissues, the lack of clinical complications following injection of twice the amount of the past largest recorded dose of allo-MSCs [38] suggests it may be clinically safe to do so, but trials with larger numbers of subject horses must be repeated to assert clinical safety.

Cell-like structures detected in vivo were occasionally smaller than measured CB-MSCs in vitro. This was expected as confocal imaging would have captured CB-MSCs in transverse, oblique, or sagittal planes as they interacted three-dimensionally with the surrounding niche, compared to the images of in vitro CB-MSCs that were flattened on the surface of the culture dish. Also, detected cell-like structures in vivo had different colocalization patterns of red and green signal compared to in vitro CB-MSCs. In vivo, red and green signal was predominantly colocalized whereas in vitro, red and green signal was typically separate although colocalization of signal was occasionally seen in some cells. The difference in signal colocalization of in vitro and in vivo CB-MSCs is likely due to differences of interactions of the cells with their environment; more specifically, signal was more commonly colocalized in in vivo cells because of three-dimensional interaction with the local niche and ligand binding, stimulating increased packaging and transportation of eGFP in PKH26 labeled organelles compared to in vitro CB-MSCs [73,74]. Regardless, there was still striking similarity between cultured CB-MSCs and in vivo cell-like structures. Hence, we considered cell-like structures that had red and green signal similarly distributed as cultured CB-MSCS to be injected fluorescently prepared CB-MSCs, irrespective of colocalization of signal.

MSC homing is defined as “the arrest of a MSC within the vasculature followed by transmigration across the endothelium” [75] and MSC engraftment is defined as integration and survival of MSCs in the target tissue after transendothelial migration [11]. Inflammation is a very important factor for influencing both homing and engraftment rates. Although not as well understood as leukocyte homing, MSCs can home to target tissues actively by rolling, adhering and migrating through vasculature activated by the inflammatory cascade, and passively by migrating through the endothelium after becoming entrapped in non-activated capillaries [75]. Recently developed techniques to increase MSC homing have integrated principles of inflammation to either “prime” MSCs to increase efficiency of transendothelial migration or modify the target tissue to be more receptive to MSC engraftment [11,75,76,77]. For example, preconditioning MSCs in a hypoxic environment will cause upregulation of CXCR4, a chemokine receptor important for homing, on MSC surface membranes and improve homing rates [11,75,76,77]. Ischemic post-conditioning of the target tissue has also been shown to improve homing and engraftment rates by decreasing reperfusion injury and creating a less hostile environment for engrafting MSCs [11]. Determining the exact mechanism of CB-MSC homing and engraftment and the relationship of the CB-MSCs with the vasculature was beyond the scope of this project design, but regardless it is widely agreed upon that inflammation plays a role in both active and passive mechanisms of MSC homing and engraftment. Therefore, we considered homing synonymous with engraftment and that homing to have occurred when red and/or green signal was detected more frequently in biopsies of wounds compared to non-wounded sites at each time period. Although only two horses were used in this pilot project and conclusions must be drawn cautiously, our preliminary results suggest CB-MSC homing and engraftment indeed occurred and temporal and regional inflammation likely influenced detection patterns.

Except for thoracic wounds on day 2, our preliminary findings suggest that CB-MSCs preferentially homed to wounds in both horses on days 1 and 2 during the acute inflammation phase, and on day 33 during the early remodeling phase, where homing and engraftment was most obvious (Table 1; Figure 8). Furthermore, the engrafted CB-MSCs in the wounded sites on day 33 were elongated and occasionally found in clusters, possibly representative of differentiation and/or proliferation (Figure 8). This is in contrast to homing and engraftment patterns of intravenously administered MSCs in mice where homing is typically greatest during the acute inflammatory phase followed by a rapid decrease or absence of engrafted MSCs as inflammation resolves [9,24,78], suggesting that the majority of engrafted MSCs are eventually rejected. Why we saw a different homing pattern is unknown, but we can speculate based on findings from other research. In a rat knee osteoarthritis model, MSCs administered intra-articularly persisted in arthritic knees more than twice as long than in normal knees and began to proliferate [79]. The investigators hypothesized that a local inflammatory environment is favorable for MSC engraftment and supports MSC viability and proliferation. In addition, other researchers have shown that successfully engrafted MSCs can recruit more locally engrafted MSCs [80] as well as local and distant endogenous MSCs [81]. Based on these studies, it is possible on day 33 that ongoing and persistent chronic inflammation in the wounds produced an inflammatory environment that was favorable for CB-MSC engraftment and proliferation, and the engrafted CB-MSCs then recruited other locally engrafted CB-MSCs to the site as well as began to proliferate. In contrast, CB-MSCs were detected only rarely if at all in biopsies of the contralateral non-wounded site because local intermittent acute inflammation created during previous biopsy collections had resolved and any previously engrafted CB-MSCs did not persist or proliferate because of lack of chronic inflammatory stimulus. Although dual Ki-67/eGFP immunohistochemistry was not performed to confirm differentiation and proliferation because it was beyond the scope of this project, we cannot disregard CB-MSC proliferation as a possible reason for increased CB-MSC detection on day 33. Regardless, the marked homing of CB-MSCs to the wounded sites and not the control sites suggests that CB-MSCs successfully homed and engrafted to the wounded site and possibly began to proliferate while continuing to recruit other engrafted CB-MSCs.

Because the original intents of this project were not to determine the relationship of the CB-MSCs with the vasculature, imaging of vessels was inconsistent and we cannot state whether the relationship of the CB-MSCs with the vasculature was consistent between biopsy sites or throughout time. However, in hindsight, the relationship of the CB-MSCs with the vasculature and apparent engraftment seemed to be related to the phases of healing and are important to note in our observations. During the inflammatory phase, CB-MSCs were detected in close relationship with the vessels, either intravascularly along the endothelium, or adjacent to the vessels within the interstitium (Figure 4 and Figure 5), but during the early remodeling phase CB-MSCs were typically detected engrafted in the interstitium and not immediately adjacent to the vessels (Figure 8). These preliminary observations are consistent with the extravasation phase and interstitial migration phase of systemic MSC homing and engraftment [75] although further studies are needed to validate this observation. 

A secondary objective was to determine whether there were differences in homing between limb and thoracic wounds. Differences between degree of CB-MSC homing in the limb and thoracic biopsies were only apparent on day 2; more specifically, mild homing was apparent in limb wounds but there were no differences in homing in thoracic wounds (Table 1; Figure 4). Having only two horses included in the study makes drawing firm conclusions difficult. However, a difference in detection of differences in homing was still anticipated as wounds on the body of horses have a more pronounced acute inflammatory response than limb wounds [1,2], which theoretically could cause greater MSC homing and recruitment and we did not observe this phenomenon. It is possible, that inflammation created at the control hemi-thorax sites from biopsy collection the day before (day 1) created an inflammatory response that caused CB-MSCs to home to the contralateral non-wounded biopsy site, negating any difference in CB-MSC homing to the thoracic wound sites on day 2. Because limb wounds have a milder inflammatory response, it is possible that confounding inflammatory recruitment of CB-MSCs was not apparent in the limbs. It is also possible that limbs have a unique feature that increases their ability to recruit MSCs compared to other areas of the body. It is widely accepted that equine limb wounds have a different temporal cytokine profile that affects wound healing compared to other areas of the body [1,2], and so it possible that limb wounds may also have unique cytokine profiles or regional anatomical features that are favorable to MSC recruitment, such as increased hypoxia-inducible factors, endothelial adhesions receptors, and signaling pathways [11,75,76]. However, because we did not detect any other differences of homing between limb and thoracic wounds at any of the other time points this theory is less likely and purely speculative although further studies investigating this theory would be interesting.

After intravenous MSC administration, the majority of MSCs are entrapped in the pulmonary vasculature, which is known as the “first pass effect” [17]. Despite this, positive effects of intravenous MSC therapy are still seen due to a rapid release of immunomodulatory factors from entrapped MSCs that systemically influence healing [17]. In addition, even low engraftment rates improve wound healing through secretion of factors that recruit endogenous MSCs and promote angiogenesis [10,81]. Even though most of our administered CB-MSCs likely became entrapped in the lungs, we still detected CB-MSCs in the biopsies, indicating that CB-MSCs can home to and engraft in cutaneous wounds after passing through the lung vasculature in horses. As far as we are aware, we are the first group to detect CB-MSCs in equine cutaneous wounds after intravenous administration. 

There are some limitations to the present study. First, this was a proof-of-principle pilot project with only two treatment horses. However, except for day 7 where horse 2 had increased homing to wounds and horse 1 did not, both horses had similar homing patterns for all time periods, and we feel our results are accurate and collectively our observations can be viewed as visual evidence of MSC homing. Second, the total fluorescence of the biopsied tissues was not quantified by software or graded. However, software evaluation can introduce further bias as subjective evaluation is required to determine background versus cellular fluorescence, while a grading system would have had limited application with only two horses. Third, the donor and the recipient horses were not cross-matched so it is possible that no adverse reactions were seen because the donors and recipients had matching haplotypes. However, domestic horses have upwards of 29 different haplotypes [82], and the CB-MSCs were pooled from five different individual donors, therefore it is very unlikely that all the donors and the recipients had matching haplotypes. Fourth, tissues and blood samples were not collected to determine if there were inflammatory responses—cellular, humoral, molecular, or otherwise. However, because the horses did not experience clinically detectable adverse reactions, cellular immune responses and immune-mediator release are likely inconsequential. Furthermore, although determining the immune response through cellular infiltration and inflammatory marker release would have been interesting, it would have limited value with only two animals in the study. Fifth, continuing to biopsy the wounds past day 33 into more advanced stages of the remodeling phase would have been ideal to determine if the CB-MSCs continued to persist or increase beyond the final biopsy day. However, persistence of engrafted CB-MSCs beyond 33 days was not anticipated based on other studies with laboratory and equine models [9,24,78], and monetary, personnel, and facility support was not available beyond the original study timeline. And finally, biopsy collection at the control sites created inflammation that may have influenced homing to the sequential control sites, although the amount of inflammation was far less than the wounded sites. This could have been controlled for by including more horses and collecting a biopsy from each horse only once at a single time period; however, this would be cost prohibitive and the benefit of using the same animal as its own control would be lost.

## 5. Conclusions

In the present proof-of-principle pilot project, we are able to report that after intravenous administration of the largest recorded dose of CB-MSCs our study horses did not have any clinical adverse effects. Furthermore, we also presented preliminary evidence of CB-MSCs preferentially homing and engrafting to wounds during the acute and early remodeling phases of cutaneous wound healing, and results suggest that there was no difference in homing between limb and thoracic wounds except on day 2 where CB-MSCs homed to limb wounds, but not thoracic wounds. The lack of adverse effects and demonstration of visual validation of homing and engraftment after intravenous administration is new to the equine literature and it is important information for further MSC research in horses. Although preliminary, the initial results are promising and future studies are warranted.

## Figures and Tables

**Figure 1 cells-09-01162-f001:**
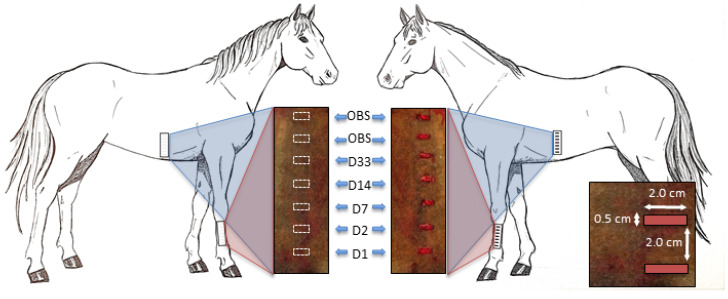
Basic schematic of wound creation and sequence of biopsy collection. On day 0, seven wounds were created on the left forelimb and hemi-thorax of each horse measuring 0.5 cm × 2.0 cm and placed 2 cm apart in a vertical orientation. Biopsies were collected on days (D) 1, 2, 7, 14, and 33 from the wound site and from the corresponding contralateral non-wounded site in a distal to proximal sequence. The top two wounds were left to heal by second intention and observed for healing characteristics. (OBS), observed.

**Figure 2 cells-09-01162-f002:**
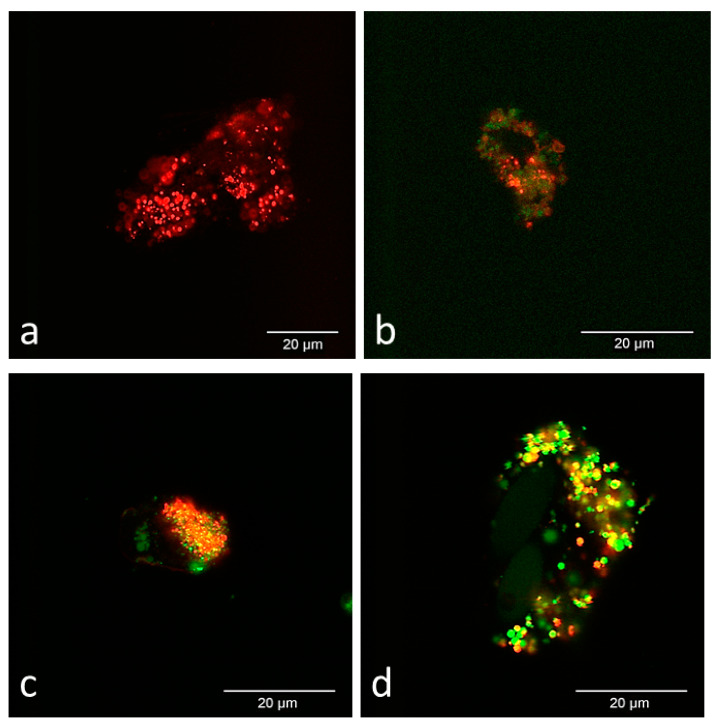
Representative confocal microscopy images of in vitro cultured prepared CB-MSCs. (**a**) Day 1, (**b**) day 7, (**c**) day 14, and (**d**) day 33. All images were captured using the same settings except for day 7 (**b**), which was captured with a higher sensitivity setting for green to enhance early subtle green signal for demonstration purposes. CB-MSCs are clumped in day 1 (**a**) and day 33 (**d**). In all images, individual CB-MSCs measured between 15–20 μm lengthwise and 10–15 μm widthwise. On day 1 (**a**), many red fluorescent dots were uniformly distributed on phospholipid membranes with very weak green signal. On day 7 (**b**), CB-MSCs maintained red signal but also began to produce stronger green signal as dots or generalized throughout the cytoplasm. On day 14 (**c**) and day 33 (**d**), CB-MSCs remained similar to day 7 (**b**) but had brighter green signal of the cytoplasm and organelles. On day 33 (**d**) green signal was more intense than previous days. Colocalization of red and green signal varied between individual cells but was typically low to moderate.

**Figure 3 cells-09-01162-f003:**
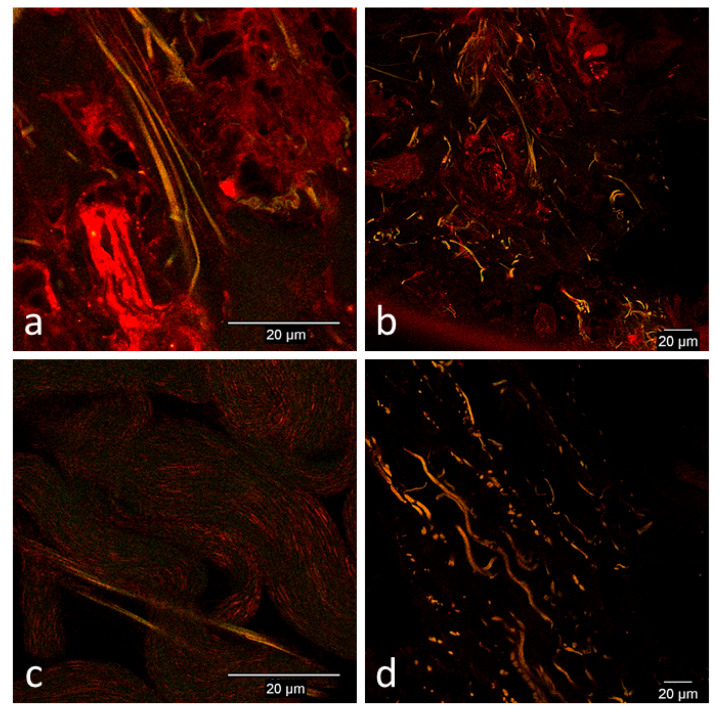
Representative confocal microscopy images of normal skin of the limb and thorax before surgical creation of wounds. **(****a**) Horse 2, thorax, (**b**) horse 2, thorax, (**c**) horse 1, thorax, and (**d**) horse 1, limb. Images were selected from either horse 1 or horse 2 and from either the limb or thorax as was determined to be the best representative image of normal skin. In both the limb and thorax, collagen fibers demonstrated normal green and red auto-fluorescence. No cell-like structures resembling cultured prepared CB-MSCs were detected.

**Figure 4 cells-09-01162-f004:**
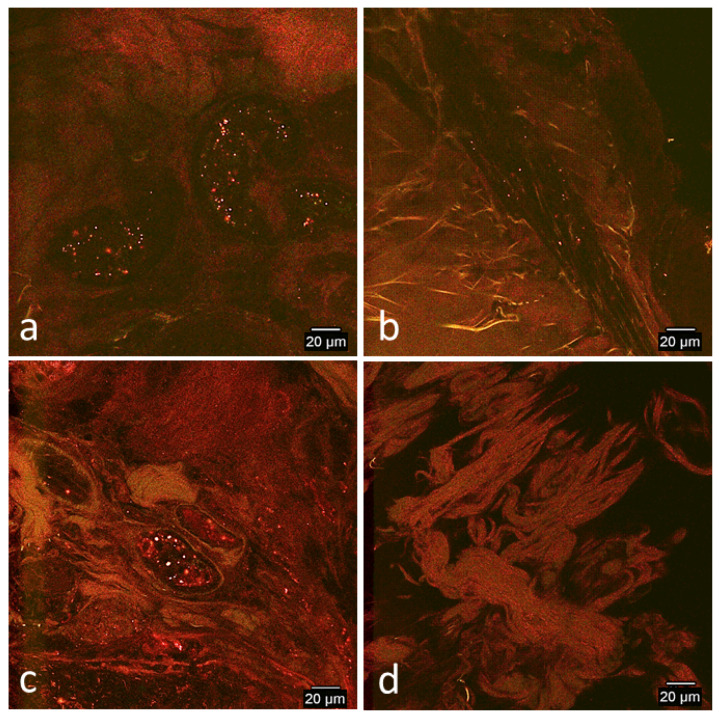
Representative confocal microscopy images of biopsies of limb and thoracic wounds and contralateral non-wounded skin on day 1. (**a**) Wounded limb, (**b**) non-wounded limb, (**c**) wounded thorax, and (**d**) non-wounded thorax. Images were selected from either horse based on best representation of patterns typical fluorescent signal patterns of both horses. Images have been selectively enhanced for purposes of demonstrating signal variation. There was slightly more homing to wounded limbs (**a**) and wounded thoraces (**c**) than the non-wounded sites (**b**,**d**). Red and green signal was highly concentrated in vasculature of wounded biopsies.

**Figure 5 cells-09-01162-f005:**
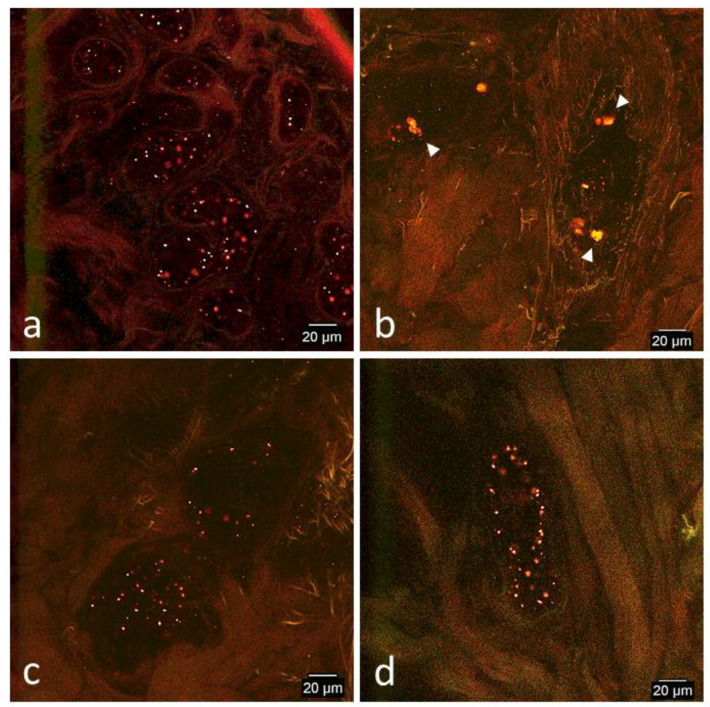
Representative confocal microscopy images of biopsies of limb and thoracic wounds and contralateral non-wounded skin on day 2. (**a**) Wounded limb, (**b**) non-wounded limb, (**c**) wounded thorax, and (**d**) non-wounded thorax. Images were selected from either horse based on best representation of patterns typical fluorescent signal patterns of both horses. Images have been selectively enhanced for purposes of demonstrating signal variation. There was slightly more homing to limb wounds (**a**) than the non-wounded limb site (**b**). There was no difference in homing between the thoracic wounds (**c**) and non-wounded thoracic sites (**d**). Signal was red and green and located within the vasculature and occasionally along the endothelium (**d**) of both wounded and non-wounded biopsies. Cell-like structures measuring 8-10 um with colocalization of red and green signal were occasionally detected adjacent to the vasculature in wounded and non-wounded limb biopsies of horse 1 (arrowheads, **b**).

**Figure 6 cells-09-01162-f006:**
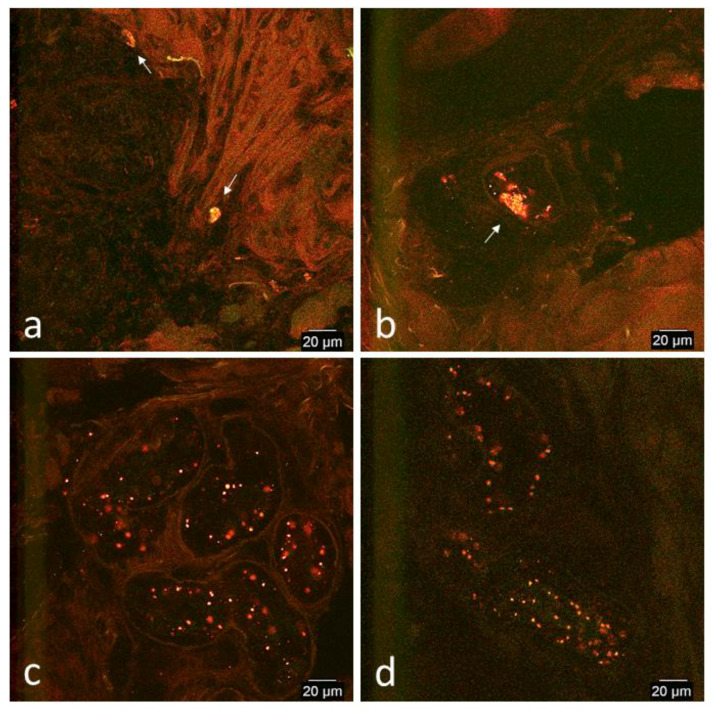
Representative confocal microscopy images of biopsies of limb and thoracic wounds and contralateral non-wounded skin on day 7. (**a**) Wounded limb, (**b**) non-wounded limb, (**c**) wounded thorax, and (**d**) non-wounded thorax. Images were selected from either horse based on best representation of patterns typical fluorescent signal patterns of both horses. Images have been selectively enhanced for purposes of demonstrating signal variation. Red and green signal was detected in all wounded and non-wounded biopsies. There was no difference in homing to wounds in either horse except for an area of the wounded thorax in horse 2 that had several cell-like structures. Signal was located within the vasculature along the endothelium in both horses (**c**,**d**) and cell-like structures consistent with in vitro CBMSCs were occasionally detected adjacent to or within the vasculature of all biopsies (arrow, **b**).

**Figure 7 cells-09-01162-f007:**
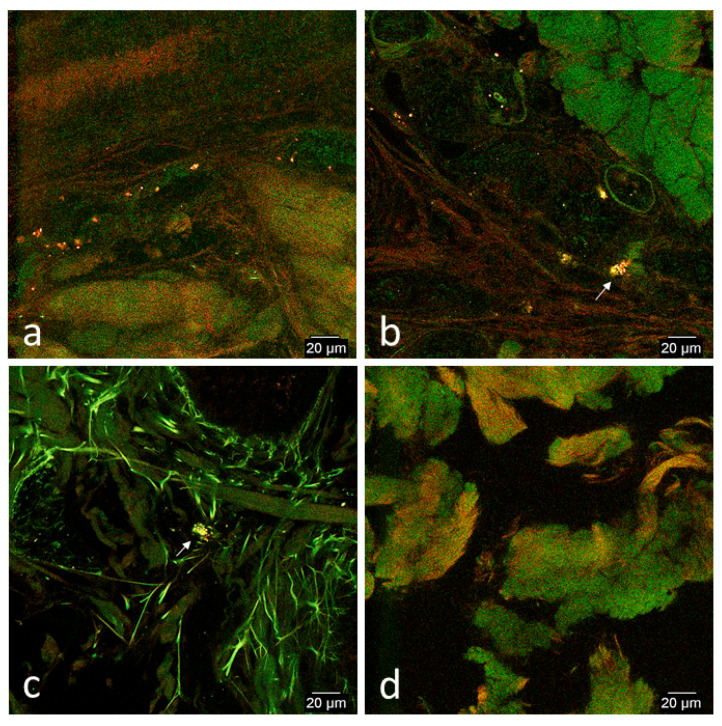
Representative confocal microscopy images of biopsies of limb and thoracic wounds and contralateral non-wounded skin on day 14. (**a**) Wounded limb, (**b**) non-wounded limb, (**c**) wounded thorax, and (**d**) non-wounded thorax. Images were selected from either horse based on best representation of patterns typical fluorescent signal patterns of both horses. Images have been selectively enhanced for purposes of demonstrating signal variation. In both horses, there was no difference in amount of distribution of red or green signal. Signal was often colocalized in small clumps and located within or just adjacent to the vasculature (**a**). A single cell-like structure consistent with in vitro CB-MSCs were detected in the interstitium of the non-wounded limb of horse 1 (arrow, **b**) and wounded thorax of horse 2 (arrow, **c**). Background auto-fluorescence was more common in all biopsies and is shown in (**d**).

**Figure 8 cells-09-01162-f008:**
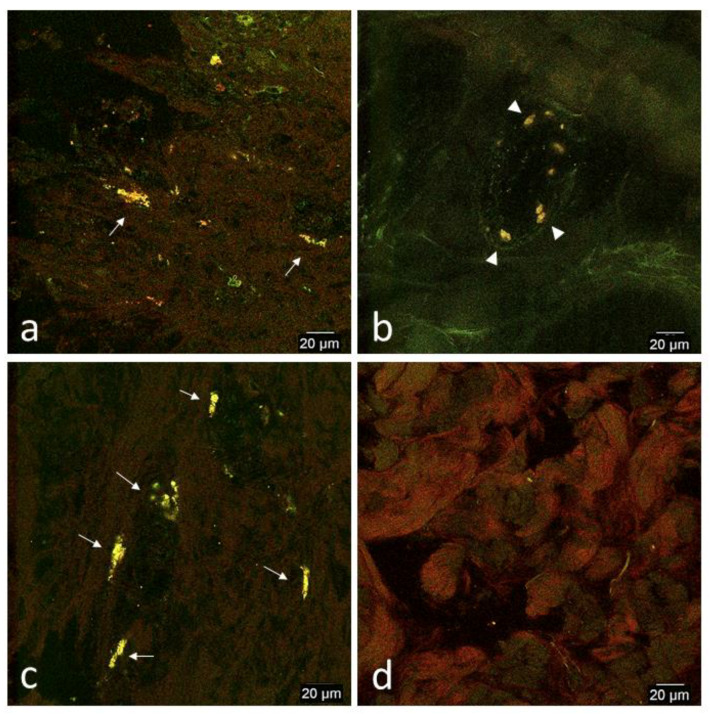
Representative confocal microscopy images of biopsies of limb and thoracic wounds and contralateral non-wounded skin on day 33. (**a**) Wounded limb, (**b**) non-wounded limb, (**c**) wounded thorax, and (**d**) non-wounded thorax. Images were selected from either horse based on best representation of patterns typical fluorescent signal patterns of both horses. Images have been selectively enhanced for purposes of demonstrating signal variation. In both horses, homing was marked in both limb and thoracic wounds and was characterized by cell-like structures consistent with in vitro CB-MSCs that appeared to be well integrated into the interstitium (arrows, **a**,**c**). In contrast, cell-like structures were only rarely detected in the non-wounded side and were rounded, located primarily in the vasculature, and were smaller than those typically detected in the wounded biopsies (arrowheads, **b**). Background auto-fluorescence was common in non-wounded biopsies and is shown in (**d**).

**Table 1 cells-09-01162-t001:** Summary of labeled CB-MSC fluorescent signal color and homing to wounds.

Biopsy Collection Day		Day 1		Day 2		Day 7		Day 14		Day 33	
Subject	Biopsy Site	Color	Homing	Color	Homing	Color	Homing	Color	Homing	Color	Homing
Horse 1	Limb Wounded	R, G	1+	R, G	1+	R, G	=	R, G	=	R, G	3+
	Limb Non-Wounded	R, G		R, G		R, G		R, G		R, G	
	Thorax Wounded	R, G	1+	R, G	=	R, G	=	R, G	=	R, G	3+
	Thorax Non-Wounded	none		R, G		R, G		R, G		R, G	
Horse 2	Limb Wounded	R, G	1+	R, G	1+	R, G	=	R, G	=	R, G	3+
	Limb Non-Wounded	R, G		R, G		R, G		R, G		R, G	
	Thorax Wounded	R, G	1+	R, G	=	R, G	=	R, G	=	R, G	3+
	Thorax Non-Wounded	R, G		R, G		R, G		R, G		R, G	

Homing was considered present if more red and/or green signal was detected in biopsies of wounds compared to the corresponding contralateral non-wounded site at the same time period, regardless of whether signal was detected intravascular or extravascular. Abbreviations: Color - (R), presence of red signal; (G), presence of green signal. Homing - (=), no difference in homing; (1+), slightly more homing; (2+), moderately more homing; (3+) markedly more homing.

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
