# Peer review of "Homing and Engraftment of Intravenously Administered Equine Cord Blood-Derived Multipotent Mesenchymal Stromal Cells to Surgically Created Cutaneous Wound in Horses: A Pilot Project"

_cells, 2020, doi:10.3390/cells9051162_

Round 1
Reviewer 1 Report
This research pursued a trifold objective:
1) determine if subject horses developed adverse reactions during or for 6 weeks post-IV administration of >1.0 x 108 fluorescently prepared allo-CB-MSCs
2) describe the presence and patterns of homing and engraftment of fluorescently prepared allo-CB-MSCs in biopsies from wounded and non-wounded cutaneous tissue of the limb and thorax during healing
and 3) determine whether preliminary results suggest limb wounds have different patterns of CB-MSC homing and engraftment than thoracic wounds over time
The research results presented are of interest to the veterinarian and scientific community. Authors demonstrated that 100 million cells administered iv to animals are tolerated without gross rejection and that these cells perform though limited in quantity and quality, homing and engraftment. This is the main asset of this paper.
While the study is well designed and done, there are several important issues that to this reviewer were missing. I will go thru the objectives and pitfalls in each one:
1) authors addressed gross disturbances in the recipient horses, i.e. gross adverse reactions, which were not found. These are good news, but why they did not look for systemic inflammatory or immunogenic markers in the hosts? I think this is a missing opportunity.
2) the second objective was partially achieved, but the results are not conclusive, two animals is much too small an n and the results are not as clear as authors claim them to be. For instances, it is hard to understand why day 7 cells in vitro express much more green label than at day 14. In figure 7 and 8, authors declare that the non wounded thorax and the wounded one had similar presence of labeling, but in the lower right image in both figures is the auto fluorescence, not the actual label of that particular sample
3) there is no answer to the third objective, again due to the small number of animals used.
The real thing is that authors labeled their manuscript as "pilot study", in that sense the manuscript is fine; however then the initial statement of the actual objectives of the research should be changed accordingly.
It is
Other minor questions:
-why the survival to PKH26 staining is so low? (50% approx). Can this be reflexed in the low number of cells found in the biopsies?
-the discussion is somewhat pretentious and is based in too many assumptions that are not proven in the research
Minor details: in the figure legends (for instances in figure 2 and 5), at least in my copy there are weird numbers in between the text, like "781" and "803 horses"
It is admirable that the authors could have used more horses and fewer injuries per horse, which would have helped reduce the effects of inflammation, however this would have gone against animal welfare and the principle of the 3 Rs.
Author Response
The authors would like to thank the reviewer for their time evaluating our manuscript and providing their valuable insight that we believe has improved its quality and contribution to literature. Please do not hesitate to let us know if you’d like to see further clarification or revisions, which we will be happy to provide. Our responses to your suggestions and queries are italicized.
This research pursued a trifold objective:
1) determine if subject horses developed adverse reactions during or for 6 weeks post-IV administration of >1.0 x 108 fluorescently prepared allo-CB-MSCs
2) describe the presence and patterns of homing and engraftment of fluorescently prepared allo-CB-MSCs in biopsies from wounded and non-wounded cutaneous tissue of the limb and thorax during healing
and 3) determine whether preliminary results suggest limb wounds have different patterns of CB-MSC homing and engraftment than thoracic wounds over time
The research results presented are of interest to the veterinarian and scientific community. Authors demonstrated that 100 million cells administered iv to animals are tolerated without gross rejection and that these cells perform though limited in quantity and quality, homing and engraftment. This is the main asset of this paper.
While the study is well designed and done, there are several important issues that to this reviewer were missing. I will go thru the objectives and pitfalls in each one:
1) authors addressed gross disturbances in the recipient horses, i.e. gross adverse reactions, which were not found. These are good news, but why they did not look for systemic inflammatory or immunogenic markers in the hosts? I think this is a missing opportunity.
Other studies have examined the immunogenic response of equids following IV administration. Williams et al. (Williams, L.B.; Co, C.; Koenig, J.B.; Tse, C.; Lindsay, E.; Koch, T.G. Response to intravenous allogeneic equine cord blood-derived mesenchymal stromal cells administered from chilled or frozen state in serum and protein-free media. Frontiers in Veterinary Science 2016, 3, 1–13.) administered 50 million allogeneic CB-MSCs IV and were able to detect stimulation of CD4+ and CD8+ lymphocytes. Knowing that allogeneic MSCs are not entirely immunoprivileged and still illicit an immune response, the greater question that we were trying to answer in this study was if there are any ill effects directly related to administration of greater number of MSCs rather than describe the molecular immunogenic response of the horse before progressing to studies with more horses with greater amounts of cells. Although we agree with the reviewer that doing serologic assays would have been interesting, they would have had added cost and have limited value in only two individuals. A comment regarding this limitation has now been put in the limitations section of the manuscript.
2) the second objective was partially achieved, but the results are not conclusive, two animals is much too small an n and the results are not as clear as authors claim them to be. For instances, it is hard to understand why day 7 cells in vitro express much more green label than at day 14. In figure 7 and 8, authors declare that the non wounded thorax and the wounded one had similar presence of labeling, but in the lower right image in both figures is the auto fluorescence, not the actual label of that particular sample
The authors agree that with two animals, making firm conclusions from the results is difficult. However, even so the authors feel that the results were consistent between the two animals and are important to report. In response though to the reviewer’s concern about conclusions of results being too pretentious, the authors have now changed the language throughout the manuscript to reinforce that results are preliminary and must be interpreted cautiously. In regards to figure 2, in the legend it is stated that the image on day 7 is enhanced to demonstrate early subtle green fluorescence that otherwise was difficult to appreciate. This has now been reworded to be more obvious to the reader. In addition, in figure 7, the authors’ intent was to show the reader the presence of the single cell-like structure found in the wounded thorax of one of the horses as a single interesting visual finding and was not typical. This has now been reworded in both the manuscript and the figure legend to be clearer to the reader. Again, in figure 8, the intent of the figure was to demonstrate to the reader the presence of cell-like structures well integrated into the interstitium in the wounded tissue and to show that although there were occasionally cell-like structures also in the non-wounded tissue, it was less common and next to the vasculature and that in fact, just background fluorescence was the more typical finding in non-wound skin. The authors have now reworded this in the manuscript and figure legend to make it clearer to the reader why certain images were chosen.
3) there is no answer to the third objective, again due to the small number of animals used.
Yes the authors agree that due to the small number of animals used that this could not be determined with certainty. The manuscript has now been reworded to acknowledge this for the reader.
The real thing is that authors labeled their manuscript as "pilot study", in that sense the manuscript is fine; however then the initial statement of the actual objectives of the research should be changed accordingly.
The authors have now altered the statement of the objectives to more strongly highlight our primary objective of determining whether there is gross rejection or adverse clinical reactions to administration of >1.00x 108 CB-MSCs and reclassify the other two objectives as “secondary” to emphasize that the results are preliminary although very interesting and worth reporting.
Other minor questions:
-why the survival to PKH26 staining is so low? (50% approx). Can this be reflexed in the low number of cells found in the biopsies?
In answering this question, we went and checked our numbers again. Although the 4.16 x 108 cells at arrival was indeed correct, that number was in fact the TOTAL cell number and not LIVE cells, as originally stated in the manuscript. The cells at arrival had a viability of 80%, resulting in 3.33 x 108 live cells. This has now been corrected on line 149 of the revised manuscript. After all in vitro manipulations including viral transfection and PKH26 staining, there were a total of 3.94 x 108 cells remaining with a cell viability of 52%, i.e., 2.04 x 108 LIVE cells and therefore each horse indeed received 1.02 x 108 live cells, as correctly reported in the original manuscript.
Given that we started with a total of 416 million cells (viability of 80%) and ended with a total of 394 million cells (viability of 52%), we had 333 million live cells to begin and after several hours of manipulation and multiple washes we had 204 million live cells. Thus ~ 38% of the reduction in viability can be attributed to the processing. Therefore, the reduction in cell viability was less than originally quoted and was due to both viral transduction and PKH26 staining, and perhaps more importantly, due to a combined 6 rounds of spinning and resuspensions. The frequent washes after each step were required to ensure proper processing and that at the end no contaminating viral vectors or PKH26 dye were remaining in the media to be transferred. Even without additional manipulations, we normally expect the cell number and viability to decrease after each washing step, especially at 400-500 x g. Since both procedures as well as multiple washings are rather harsh on cells, the reduced numbers are not unexpected and in fact reasonable.
From our previous experience with PKH26 and the same adeno-associated viral vectors (Honaramooz et al., 2002; 2003; 2008, Zeng et al., 2013), we believe once the cells survive the initial in vitro manipulations and are either cultured or transplanted, their survival rate remains high. We have been able to document presence of live PKH26-labled germ cells up to 12 weeks after in vivo transfer. Although it is difficult to know the exact numbers, we believe a larger loss of cells would be upon injection due the first pass effect, which is already discussed in the manuscript. Moreover, no matter how many cells ended up being available to the tissues, given the very small ratio of the biopsies to the whole body of a horse, it was actually surprising to see any number of cells.
-the discussion is somewhat pretentious and is based in too many assumptions that are not proven in the research
The authors have now altered the language throughout the manuscript to emphasize that the results are preliminary and only include two animals, yet are interesting to report and speculate. If there are any specific statements or areas within the manuscript that the reviewer would like to be revised more thoroughly please identify to the authors specifically where in the manuscript.
Minor details: in the figure legends (for instances in figure 2 and 5), at least in my copy there are weird numbers in between the text, like "781" and "803 horses"
The authors apologize for the oversight and the typos have now been corrected.
It is admirable that the authors could have used more horses and fewer injuries per horse, which would have helped reduce the effects of inflammation, however this would have gone against animal welfare and the principle of the 3 Rs.
Yes, the authors acknowledge that having one horse only for each created wound and biopsy collection time period would have been ideal to decrease confounding inflammation created by previous biopsy collection. However, due to expenses of procuring the cells and housing the horses, this would have been cost prohibitive, not to mention increased animal use rather than reducing the number of animals needed for this proof-of-principle study. This has now been acknowledged in the limitations section.
Reviewer 2 Report
The manuscript by Mund et al. reports a pilot study of systemic (IV) injection of allogeneic cord blood-derived MSCs into two horses to treat cutaneous wound in the limb and thorax. Authors report the safety of the procedure using a high number of cells and show MSC homing to injury sites and subsequent engraftment through fluorescence of the cells. The study is very preliminary and no quantitative data is provided. However, studies with large animals always have different implications in terms of budget, ethics and trained personel, to mention a few, when compared to small animal models. Main limitations of the study are very well discussed at the end of the manuscript and the rationale is clearly identified throughout the manuscript. Nevertheless, some issues were overlooked throughout the manuscript, as highlighted in detail below.
- Introduction, Line 66, “immunosuppressive” properties of MSCs are debatable. Immunomodulation is the most commonly accepted MSC feature.
- In figure 2, only one cell is seen in each image. Could authors provide also a lower magnification showing a higher number of cells? Can authors quantify the red and green fluorescence overtime in culture? As well as co-localization?
- Results provided on figure 3 are not clear. What is the rationale? Also, figure caption seems to be missing the limb of horse 2. What do authors mean by cell-like structures? Corresponding section of the text is also confusing, as this is not clear.
- No comments on tissue healing are added to the study. This could help understanding the benefits of CB-MSCs in wound healing for these cases.
- Inflammatory markers and immune cell infiltration should have been determined and could add value to the study.
- A schematic timeline of the proposed mechanism could also clarify the message of the study.
Despite the obvious limitations of the study, all the biological (microscopy) data and the absence of quantitative results, authors discuss their work in a clear, to-the-point, concise and effective way. I would like to congratulate the authors for their job on discussing the study, carefully highlighting all the limitations of the study and justifications. This is rarely seen nowadays in Science and our job is to raise new questions, not to answer them all.
Author Response
The authors would like to thank the reviewer for their time evaluating our manuscript and providing their valuable insight that we believe has improved its quality and contribution to literature. Please do not hesitate to let us know if you’d like to see further clarification or revisions, which we will be happy to provide. Our responses to your suggestions and queries are italicized.
The manuscript by Mund et al. reports a pilot study of systemic (IV) injection of allogeneic cord blood-derived MSCs into two horses to treat cutaneous wound in the limb and thorax. Authors report the safety of the procedure using a high number of cells and show MSC homing to injury sites and subsequent engraftment through fluorescence of the cells. The study is very preliminary and no quantitative data is provided. However, studies with large animals always have different implications in terms of budget, ethics and trained personel, to mention a few, when compared to small animal models. Main limitations of the study are very well discussed at the end of the manuscript and the rationale is clearly identified throughout the manuscript. Nevertheless, some issues were overlooked throughout the manuscript, as highlighted in detail below.
Introduction, Line 66, “immunosuppressive” properties of MSCs are debatable. Immunomodulation is the most commonly accepted MSC feature.
The authors would like to direct the reviewer to the source cited in Line 66 (Macrin, D.; Joseph, J.P.; Pillai, A.A.; Devi, A. Eminent sources of adult mesenchymal stem cells and their therapeutic imminence. Stem Cell Reviews and Reports 2017, 13, 741–756). Evidence is provided in many studies in Macrin’s review of immunosuppressive abilities of MSCs. However, immunosuppression can be considered a type of immunomodulation, thus the authors will revise the manuscript to refer to changes in the immune response as “modulated” rather than “suppressed”.
In figure 2, only one cell is seen in each image. Could authors provide also a lower magnification showing a higher number of cells? Can authors quantify the red and green fluorescence overtime in culture? As well as co-localization?
The images in a) and d) show several cells clumped together as stated in the figure legend. The reference bar of 20um is the approximate length of a single cell. These images were chosen as they showed they had the best image clarity and the authors fear that the clarity will be lost if lower magnification was provided as well as there are no other cells in the field of view of the selected cells at lower magnification. We agree that if planned ahead, quantifying fluorescence and co-localization over time could have provided information about the behaviour of in vitro cultured cells. However, this would have required taking sufficient representative images with regular intervals which was not planned. Rather, we considered imaging of in vitro cells only as a qualitative added assurance of expression of GFP and persistence of the PKH26 dye over time and not as a quantitative endpoint. Additionally, given the vast differences between the in vitro and in vivo situation, we are not sure how valuable such an in vitro analysis would have been in serving as a representative control for the behaviour or fate of the cells in vivo.
Results provided on figure 3 are not clear. What is the rationale? Also, figure caption seems to be missing the limb of horse 2. What do authors mean by cell-like structures? Corresponding section of the text is also confusing, as this is not clear.
The rationale of figure 3 was to provide the best representative images of either the limb or thorax of either horse 1 or horse 2 to demonstrate what normal background fluorescence looked like in skin biopsies before wound creation and administration of cells, and that there were no cells that resembled the fluorescent cultured cells that could be mistaken for prepared MSCs later on. The term “cell-like structure” is in reference to cells that have fluorescence similar to cultured fluorescing MSCs in figure 2. The authors understand how this may be confusing to the reviewer and potential readers and have now changed the description to make this clearer. Please let us know if further clarification is needed.
No comments on tissue healing are added to the study. This could help understanding the benefits of CB-MSCs in wound healing for these cases.
The authors agree that commenting specifically on how MSC therapy may help equine wound healing specifically would add value to the paper. A paragraph has now been added at the beginning of the discussion about how horse limb wounds and human chronic wounds have similarities and how MSC therapy may improve wound healing in horses.
Inflammatory markers and immune cell infiltration should have been determined and could add value to the study.
Other studies have examined the immunogenic response of equids following IV administration. For instance, Williams et al. (Williams, L.B.; Co, C.; Koenig, J.B.; Tse, C.; Lindsay, E.; Koch, T.G. Response to intravenous allogeneic equine cord blood-derived mesenchymal stromal cells administered from chilled or frozen state in serum and protein-free media. Frontiers in Veterinary Science 2016, 3, 1–13.) administered 50 million allogeneic CB-MSCs IV and were able to detect stimulation of CD4+ and CD8+ lymphocytes. Knowing that allogeneic MSCs are not entirely immunoprivileged and still illicit an immune response, the greater question that we were trying to answer in this study was if there are any ill effects directly related to administration of greater number of MSCs rather than describe the molecular immunogenic response of the horse before progressing to studies with more horses with greater numbers of cells. Although we agree with the reviewer that doing serologic assays would have been interesting, they would have had added cost and have limited value in only two individuals. A comment regarding this limitation has now been put in the limitations section of the manuscript.
A schematic timeline of the proposed mechanism could also clarify the message of the study.
The authors agree that a visual schematic of the proposed mechanism would be advantageous and a nice addition to the manuscript. However, as the intents of this pilot project were a proof-of-principle, the mechanisms of homing and engraftment were not directly investigated. The authors feel that a visual schematic would be more appropriate in a review paper where we can compare our findings of this paper with other studies; currently the authors are preparing a separate review manuscript which will likely have a schematic as the reviewer has suggested.
Despite the obvious limitations of the study, all the biological (microscopy) data and the absence of quantitative results, authors discuss their work in a clear, to the point, concise and effective way. I would like to congratulate the authors for their job on discussing the study, carefully highlighting all the limitations of the study and justifications. This is rarely seen nowadays in Science and our job is to raise new questions, not to answer them all.
Thank you. This is a very kind and much appreciated comment.
Reviewer 3 Report
The authors here described the use of allogenic cord blood-derived mesenchymal stem cells for wound healing in horses. Two horses were used as proof of the principle pilot project. The cells were tested for homing, engrafting and safety in the horses. While the study has clinical implications in large animals and humans there are few concerns that should be addressed.
1) No characterization of CB-MSCs. Immunostaining or FACS data with MSC markers should be provided.
2) Similarly, there is no in vivo characterization data of MSCs after 33 days. How do you know they are still MSCs after 33 days in vivo and did not dedifferentiate into any other cells. The efficacy of the method relies on this important piece of data.
Author Response
The authors would like to thank the reviewer for their time evaluating our manuscript and providing their valuable insight that we believe has improved its quality and contribution to literature. Please do not hesitate to let us know if you’d like to see further clarification or revisions, which we will be happy to provide. Our responses to your suggestions and queries are italicized.
The authors here described the use of allogenic cord blood-derived mesenchymal stem cells for wound healing in horses. Two horses were used as proof of the principle pilot project. The cells were tested for homing, engrafting and safety in the horses. While the study has clinical implications in large animals and humans there are few concerns that should be addressed.
1) No characterization of CB-MSCs. Immunostaining or FACS data with MSC markers should be provided.
The cells that were provided for this study were from the cell lines that were previously characterized as having surface marker phenotypes consistent with MSCs (Tessier, L.; Bienzle, D.; Williams, L.B.; Koch, T.G. Phenotypic and immunomodulatory properties of equine cord blood-derived mesenchymal stromal cells. PLoS ONE 2015, 10, 1–20.) The authors have now added “…characterized and…” to line 139 to make this more clear to the reviewer and readers.
2) Similarly, there is no in vivo characterization data of MSCs after 33 days. How do you know they are still MSCs after 33 days in vivo and did not dedifferentiate into any other cells. The efficacy of the method relies on this important piece of data.
The provider of the cells had previously determined that the cells were multipotent and capable of differentiating into adipogenic, osteogenic, and chondrogenic cell lineages in vitro (Koch, T.G.; Thomsen, P.D.; Betts, D.H. Improved isolation protocol for equine cord blood-derived mesenchymal stromal cells. Cytotherapy 2009, 11, 443–447). Although it is indeed possible that the cells had differentiated into different cell types in vivo while maintaining fluorescence, the main intent of this pilot project was to evaluate the feasibility of administering this high number of cells and determine the clinical response of the horses - the presence of our cells at day 33 was unanticipated and surpassed our expectations. Thus, characterizing the cells at day 33 was beyond the scope of this initial project although in future projects the authors will anticipate their presence and be prepared to characterize them further. Furthermore, the authors respectfully partially disagree with the reviewer that the efficacy of IV MSC therapy should not rely only on the possibility of cell differentiation and/or replication, but also a clinically measurable improvement in wound healing when compared to controls, although in vivo characterization would help elucidate the mechanism in future studies. A comment has been added in line 140 to make clear to the reader that the cells also had trilineage capabilities.
Round 2
Reviewer 3 Report
The authors have now clarified the reviewer's concern.
Author Response
Great. Thank you for time and expertise. It is much appreciated.